# Effect of Dietary *Fructus mume* and *Scutellaria baicalensis* Georgi on the Fecal Microbiota and Its Correlation with Apparent Nutrient Digestibility in Weaned Piglets

**DOI:** 10.3390/ani12182418

**Published:** 2022-09-14

**Authors:** Feng Zhang, Erhui Jin, Xiaodan Liu, Xu Ji, Hong Hu

**Affiliations:** 1College of Animal Science, Anhui Science and Technology University, Chuzhou 233100, China; 2Anhui Province Key Laboratory of Animal Nutrition Regulation and Health, Chuzhou 233100, China; 3Anhui AnFengT Animal Medicine Industry Co., Ltd., Hefei 230031, China; 4Anhui Province Key Laboratory of Livestock and Poultry Product Safety Engineering, Institute of Animal Science and Veterinary Medicine, Anhui Academy of Agricultural Sciences, Hefei 230031, China

**Keywords:** traditional Chinese medicine, weaned piglets, apparent nutrient digestibility, gut microbiota

## Abstract

**Simple Summary:**

Traditional Chinese medicine (TCM) is based on ancient Chinese medical principles. In China, these medicines have played a marked role in treating various diseases and maintaining human health for thousands of years. TCM is also increasingly considered a potential alternative to the use of antibiotics in pig production and has attracted a great deal of research interest because it is simple, convenient, cheap, and effective. However, there are few studies on the effects of dietary TCM supplementation on the gut microbiota and the apparent nutrient digestibility of weaned piglets. In our study, dietary *Fructus mume* and *Scutellaria baicalensis* Georgi improved growth performance and increased the apparent ether extract (EE) digestibility by modulating gut microbial composition and structure, favoring the health of weaned piglets.

**Abstract:**

Traditional Chinese medicine (TCM) has long been demonstrated to exert a therapeutic effect on various diseases and has been used as a substitute for antibiotics in pig production. However, few studies have investigated the relationship between the intestinal microbiota and apparent nutrient digestibility when weaned piglet diets are supplemented with TCM. One hundred and sixty-two 25-day-old weaning piglets were housed in an environmentally controlled nursery facility and fed a basal diet (control group, *n* = 54) or a TCM complex (*Fructus mume* 1%, *Scutellaria baicalensis* Georgi 3%) (TCM group, *n* = 54), or a fermented diet with a complex of these two TCMs (F-TCM group, *n* = 54). Compared with the control group, in the TCM and F-TCM groups, the average daily gain (ADG) increased (*p* < 0.05), the F:G ratio and diarrhea rate decreased (*p* < 0.05), and the apparent digestibility of dry matter (DM) and ether extract (EE) of weaned piglets increased (*p* < 0.05). *Bacteroidetes* and *Firmicutes* were the predominant phyla, representing approximately 95% of all sequences. The abundance of four genera and 10 OTUs (belonging to *Ruminococcaceae*_UCG-014, *Lachnoclostridium*, *Prevotellaceae*_NK3B31 group, *Prevotella*_1) were negatively correlated with apparent EE digestibility (*p* < 0.05). The results suggest that weaned piglets fed with antibiotic-free diets supplemented with *Fructus mume* and *Scutellaria baicalensis* Georgi gained more weight and were healthier. When added to the diet, the complex of these two TCMs may have a direct impact on apparent EE digestibility by modifying the gut microbial composition, which favors the health of weaned piglets.

## 1. Introduction

Traditional Chinese medicine (TCM) is used under the guidance of ancient Chinese medicinal philosophies [1]. TCM has played a marked role in disease prevention and health improvement and has been investigated for thousands of years in China [2,3,4]. With a desperate worldwide need to reduce antibiotic usage in human and veterinary medicine, research into the therapeutic effects of TCM has attracted much attention in recent years. TCM has been demonstrated to exert a therapeutic effect on various diseases, such as diabetes [2], hypertension [5], gastric cancer [6], ulcerative colitis [7], colorectal cancer [8], etc. Knowledge about the underlying pharmacological mechanisms of TCM is still scarce.

It has been suggested that the therapeutic effect of TCM is closely related to the gut microbiota [1,9], which is the bridge between the body and the external environment, as there is a reciprocal link between the two. On the one hand, the improvement produced by the pharmacological activity of TCM depends on the gut microbiota [6,7]. The gut microbiota can promote the transformation and metabolism of TCM components by metabolizing TCM into specific molecules, such as alkaloids, flavonoids, and polysaccharides, which are easily absorbed in the intestine [2,6]. For example, paeoniflorin can be catalyzed into paeoniflorgenin and paeoniflorin by *Lactobacillus brevis* and *Bacteroides fragilis*, and puerarin is converted into daidzein by *Bifidobacterium* and *E. faecalis*, and aconitine can be decomposed into lipoaconitine by *Clostridium butyricum* and *B. fragilis* [6]. On the other hand, when present in the digestive tract, TCM can promote the growth of probiotic bacteria and inhibit pathogens, as well as prevent bacterial transmission, thus regulating the microenvironment and maintaining the balance of the microflora [6,7]. For example, flavonoids, polysaccharides, and saponins in TCM serve as prebiotics that regenerates the gut microbiota. *Escherichia coli* can be directly inhibited by cinnamon essential oil [8]. The omics technologies, such as microbiomics and metabolomics, have been considered pivotal tools to help us understand the underlying mechanisms between TCM and gut microbiota.

Apparent nutrient digestibility indicates the digestibility of feed ingredients by animals and is also an important indicator used to evaluate the nutritional value of feed ingredients in diets. Previous studies had reported that dietary supplementation with TCM improved the growth performance of heat-stressed beef cattle by increasing the apparent digestibility of organic matter (OM), crude protein (CP), and acid detergent fiber (ADF) when TCM plus γ-aminobutyric acid (GABA) was used in the diet, and the apparent EE and neutral detergent fiber (NDF) digestibility also increased [10,11]. Furthermore, in lambs and hogs, dietary TCM increased the apparent digestibility of DM, OM, CP, and NDF [12]. The research results cited above may suggest that TCM supplementation of the diet would be favorable for improving the apparent nutrient digestibility of feed. Similar studies on pig models have rarely been reported.

During weaning, piglets have to face a number of challenges that are critical and stressful [13]. On the one hand, to accelerate the pace of banning the use of antibiotic growth promoters (AGPs) in China, the addition of TCM to the feed has been considered a substitute for antibiotics as a feed additive [14]. On the other hand, physiological characteristics after weaning indicate that the weaned piglets may have unique intestinal microflora [15]. Thus, at this particular stage, studies on the effect of dietary TCM on the gut microbiota and the apparent nutrient digestibility of weaned piglets have an important scientific value. Studies unraveling the relationship between the two related research strands have not been reported so far.

The present study was conducted in two stages. First, TCMs with potent antibacterial properties were selected for further application in the production of fermented feed. Second, the growth performance, apparent nutrient digestibility, and fecal microbiota of weaned piglets were investigated. This study focused on the correlation between gut microbial communities and apparent nutrient digestibility in piglets.

## 2. Materials and Methods

### 2.1. Antibacterial Susceptibility Testing

The antibacterial properties of eight TCMs, namely *Fructus mume*, *Scutellaria baicalensis* Georgi, *Rhizoma imperatae*, *Paeoniae radix alba*, *Plantaginis semen*, *Eclipta prostrata*, *Fructus arctii*, and *Portulaca oleracea* L., were determined for both Escherichia coli and Salmonella isolates. All the minimum inhibitory concentration (MIC) tests were conducted twice in order to ensure that the results were representative. The MICs of selected TCMs were determined by the agar dilution method [16,17]. The methods and standards followed the relevant regulations of the Clinical and Laboratory Standards Institute (CLSI) [18]. The double dilution method was used to dilute the eight TCM infusions to the required concentration gradients, then sterile Mueller–Hinton (MH) agar was added and mixed to prepare the agar plates, and the bacterial suspensions (Escherichia coli and Salmonella) with a turbidity of 0.5 MCF were diluted and inoculated on the MH agar plates [17] in an inverted culture for 16–18 h at 37 °C. The MICs of the Chinese medicines were recorded when the plates showed no bacterial growth.

### 2.2. Preparation of Fermented MIXED Feed

The *Bacillus subtilis*, *Lactobacillus,* and *Saccharomyces* strains used in the present experiment were isolated and generously given by the College of Food Science and Technology, Guangdong Ocean University. A 300 g basal substrate including corn and soybean meal (mass ratio 3:1), as well as TCMs with potent antibacterial properties (*Fructus mume* or *Scutellaria baicalensis* Georgi in proportions of 1%, 3%, and 10%), was mixed and supplemented with sterile water to achieve a 60% moisture content. The mixed substrate was divided into two treatment parts: one part was inoculated with *B. subtilis*, *Lactobacillus,* and *Saccharomyces* (a proportion of 0.2% each, 10^8^ cfu/g) and then transferred to a plastic bag, while the other part of the mixed substrate was directly transferred to a plastic bag; all plastic bags were sealed and incubated at 37 ℃ for 144 h. The composition of the fermented mixed feeds is shown in Table 1.

### 2.3. Fermented Mixed Feed Parameters

Each of the 14 mixed fermentation treatments was prepared in 6 repeated plastic bags, corresponding to a 6-day (144 h) fermentation. The pH value was recorded every 24 h, and a digital pH meter was used to measure the pH of samples after calibration with standard buffers (pH 4.0 and 7.0). The results of pH values at different fermentation times is shown in Appendix A. Miscellaneous bacteria identification and antibacterial susceptibility tests were performed every 24 h.

### 2.4. Animals, Diets, and Experimental Design

This study was approved by and implemented under the supervision of the guidelines for the care and use of experimental animals of the Ministry of Science and Technology of the People’s Republic of China (Approval Number: 2006-398). The experimental protocols were approved by the experimental Animal Ethical Committee of Anhui Science and Technology University.

In total, 162 crossbred (Duroc × Yorkshire) weaning piglets (weaned at 25 days of age) with an initial average body weight (BW) of 7.55 ± 0.47 kg were selected for the 30-day experiment. All the piglets were randomly assigned to 3 dietary groups with 3 replications per treatment and 18 pigs per pen. The BW and sex were balanced among each treatment as follows: (1) piglets in the control group fed the basal diet (control group; *n* = 54); (2) piglets in the treatment group fed a diet with added selected TCMs with potent antibacterial properties (*Fructus mume* 1%, and *Scutellaria baicalensis* Georgi 3%) added to the basal diet (TCM group; *n* = 54); (3) piglets in the fermentation treatment group fed a diet supplemented with a selected TCM complex (*Fructus mume* 1%, *Scutellaria baicalensis* Georgi 3%), fermentation strains (*B. subtilis*, *Lactobacillus,* and *Saccharomyces*; 0.2% each, 10^8^ cfu/g), and sterile water, fermented at 37 °C for 144 h (F-TCM group; *n* = 54) (Table 2). All corn- and soybean-based diets had no antibiotics and conformed to the nutrient requirements of the US National Research Council [19]. Environmentally controlled nursery facilities with slatted plastic flooring and mechanical ventilation were used to house the pigs. The pre-feeding period of the weaned piglets was as follows: from 25 to 30 days of age, all piglets were fed with the basal diet, and the formal feeding trial was from 30 to 60 days of age. All pigs were fed twice a day individually (at 06:00 and 18:00) and allowed free access to feed and ad libitum water during the entire experimental period. There were no antibiotics in the feed or administered therapeutically to the pigs.

### 2.5. Determination of the Growth Performance and Diarrhea Rate

The individual BW of pigs with empty stomachs was measured at 05:00 on Days 1, 15, and 30 of the experimental period; the feed offered and residual feed were weighed and recorded daily and used to determine the ADG, average daily feed intake (ADFI), and feed-to-gain (F:G) [11]. The incidence and severity of piglet diarrhea were assessed by fecal consistency. If the piglets had moderately fluid feces and frothy diarrhea, in which the feces were definitely unformed and very watery, they were considered to be diarrheic [20]. The diarrhea rate of piglets was recorded daily and calculated as follows: diarrhea rate (%) = the number of pigs with diarrhea × diarrhea days/(total number of pigs × experimental days) × 100%, where the number of pigs with diarrhea was defined as the number of piglets with diarrhea observed each day [21].

### 2.6. Assessment of Apparent Nutrient Digestibility

According to the method described by Fouhse et al. [22] and Niu et al. [15], samples of feed were collected daily during the experimental period, and the total daily feces from each pig were collected on the last 3 days. The collected feces for each piglet were composited and mixed thoroughly; approximately 100 g of feces was subsampled after thorough mixing, and all feed and feces samples were dried at 65 °C to constant weight for subsequent analysis.

Acid-insoluble ash (AIA) was used as an indigestible marker to assess the digestibility of the dietary components according to the procedure of the Association of Official Analytical Chemists (AOAC 942.05) [23]. The DM was analyzed according to the procedures described by Xie et al. [24]. The EE content was measured using the Soxhlet extraction method (AOAC 920.85), which was performed with a Soxhlet apparatus [15]. The CP content was measured via the Kjeldahl method (AOAC 984.13) using a Kjeltec 8400 analyzer unit (Foss, Beijing, China). CF analysis was carried out using the ANKOM A200 filter bag technique (AOAC 962.09) [15]. The ingredient composition and nutrient specifications of the basal and experimental diets were calculated as previously reported, and the apparent nutrient digestibility was calculated by the following equation: $\mathrm{apparent}\;\mathrm{nutrient}\;\mathrm{digestibility}=\frac{\left(\mathrm{nutrient}/\mathrm{AIA}\right)\mathrm{diet}-\left(\mathrm{nutrient}/\mathrm{AIA}\right)\mathrm{digesta}}{\left(\mathrm{nutrient}/\mathrm{AIA}\right)\mathrm{diet}}$ [15,25,26].

### 2.7. Fecal Sample Collection, DNA Extraction, 16S rRNA Gene Amplification, and Illumina HiSeq 2500 Sequencing

The fecal samples were collected on the last day of the experiment from 18 pigs, where two pigs were selected randomly within each pen (one male and one female). The feces were stored at −80 ℃ for subsequent analyses.

The methods in this section are similar to those used in our previous studies [27,28], in which the fecal microbial DNA was extracted using HiPure Stool DNA Kits (Magen, Guangzhou, China) according to the manufacturer’s protocols. The 16S rDNA V3–V4 region of the ribosomal RNA gene was amplified by PCR (95 °C for 2 min, followed by 27 cycles at 98 °C for 10 s, 62 °C for 30 s, and 68 °C for 30 s, with a final extension at 68 °C for 10 min) using the primers 341F: CCTACGGGNGGCWGCAG and 806R: GGACTACHVGGGTATCTAAT, where the barcode is an eight-base sequence unique to each sample. PCR reactions were performed in triplicate in 50 μL mixtures containing 5 μL of 10 × KOD Buffer, 5 μL of 2.5 mM dNTPs, 1.5 μL of each primer (5 μM), 1 μL of KOD polymerase, and 100 ng of the template DNA.

Amplicons were extracted from 2% agarose gels and purified using the AxyPrep DNA Gel Extraction Kit (Axygen Biosciences, Union City, CA, USA) according to the manufacturer’s instructions and quantified using an ABI StepOnePlus Real-Time PCR System (Life Technologies, Foster City, CA, USA). Purified amplicons were pooled in an equimolar mixture and paired-end sequenced (2 × 250) on an Illumina HiSeq platform according to the standard protocols. The raw reads were deposited into the NCBI Sequence Read Archive (SRA) database (accession number: SRR9566631).

### 2.8. Bioinformatics Analysis

#### 2.8.1. Read Filtering

Raw data containing adapters or low-quality reads would affect the subsequent assembly and analysis. Thus, in order to obtain high-quality clean reads, the raw reads were further filtered according to the following rules using FASTP (https://github.com/OpenGene/fastp, accessed on 14 September 2022): (1) removing reads containing more than 10% of unknown nucleotides (N), (2) removing reads containing less than 60% of bases with a quality (Q-value) > 20.

#### 2.8.2. Read Assembly

Paired-end clean reads were merged as raw tags using FLSAH [29] (version 1.2.11) with a minimum overlap of 10 bp and mismatch error rates of 2%.

#### 2.8.3. Raw Tag Filtering

Noisy sequences of raw tags were filtered by the QIIME [30] (version 1.9.1) pipeline under specific filtering conditions [31] to obtain high-quality clean tags.

#### 2.8.4. Chimera Checking and Removal

Clean tags were searched against the reference database (http://drive5.com/uchime/uchime_download.html, accessed on 14 September 2022) to perform reference-based chimera checking using the UCHIME algorithm (http://www.drive5.com/usearch/manual/uchime_algo.html, accessed on 14 September 2022). All chimeric tags were removed, and the effective tags finally obtained were used for further analysis.

#### 2.8.5. OTU Cluster

The effective tags were clustered into operational taxonomic units (OTUs) of ≥ 97% similarity using the UPARSE [32] pipeline. The tag sequence with the highest abundance was selected as a representative sequence within each cluster. Between-group Venn analysis was performed in R (version 3.4.1, https://cran.r-project.org/bin/windows/base/old/3.4.1/, accessed on 14 September 2022) to identify unique and common OTUs.

#### 2.8.6. Taxonomic Classification

The representative sequences were classified into organisms by a naive Bayesian model using an RDP classifier [33] (version 2.2, http://rdp.cme.msu.edu/, accessed on 14 September 2022) based on the SILVA [34] database (https://www.arb-silva.de/, accessed on 14 September 2022), with the confidence threshold values ranging from 0.8 to 1. The abundance statistics of each taxon were visualized using Krona [35] (version 2.6, https://github.com/marbl/Krona/releases/tag/v2.6.1, accessed on 14 September 2022). Biomarker features in each group were screened by Metastats [36] (version 20090414) and LEfSe software [37] (version 1.0, https://github.com/waldronlab/lefser, accessed on 14 September 2022).

#### 2.8.7. Alpha Diversity Analysis

Chao1, Simpson, and all other alpha diversity indexes were calculated in QIIME. OTU rarefaction curves and rank abundance curves were plotted in QIIME. An alpha index comparison between groups was calculated by Welch’s *t*-test and Wilcoxon’s rank test. The alpha index comparison among the groups was computed by Tukey’s HSD test and the Kruskal–Wallis H-test.

### 2.9. Statistical Analysis

Data were compared among the groups using a one-way ANOVA test after normal test processing and conversion if necessary. The relative abundance of microbial communities in feces and data that did not follow a normal distribution were processed using the nonparametric Kruskal–Wallis test. Correlation analysis was performed by Pearson’s correlation tests. Significant differences were considered at *p* < 0.05. The initial body weight was included as a covariate in the growth performance analysis. The statistical analyses were conducted using SPSS Statistics (Version 22, https://www.ibm.com/analytics/spss-statistics-software, accessed on 14 September 2022) [27,28].

## 3. Results

### 3.1. Antibacterial Characteristics

To evaluate the antibacterial characteristics of eight TCMs, we analyzed the MICs of eight TCMs against *Escherichia coli* and *Salmonella* (Table 3). All eight TCMs had antibacterial effects on *Escherichia coli* and *Salmonella*. Compared with other TCMs, *Fructus mume* and *Scutellaria baicalensis* Georgi showed potent antibacterial properties and were then used in the subsequent preparation of fermented mixed feed.

### 3.2. Growth Performance

The ADG, ADFI, and the F:G ratio of the experimental piglets were measured to assess their growth performance (Table 4). The ADG of the TCM and F-TCM groups increased significantly by 51.3% (*p* < 0.05) and 53.2% (*p* < 0.05) compared with the control group, respectively. Compared with the control group, the F:G ratio of the TCM and F-TCM groups decreased significantly by 33.5% (*p* < 0.05) and 26.2% (*p* < 0.05), respectively.

### 3.3. Diarrhea Rate

The diarrhea rate of piglets was determined according to the literature [21] (Table 5). Compared with the F-TCM group, the DM of feces decreased in control and TCM groups (*p* < 0.05). Compared with the control group, the diarrhea rate of piglets in the TCM and F-TCM groups was significantly decreased by 41.7% (*p* < 0.05) and 65.6% (*p* < 0.05).

### 3.4. Apparent Nutrient Digestibility

To determine whether TCM supplementation could improve the apparent nutrient digestibility in weaned piglets, we assessed the digestibility of DM, CP, EE, CF, ash, and NFE (Table 6). The apparent digestibility of DM and EE increased in the TCM and F-TCM groups in comparison with the control group (*p* < 0.05), while no significant difference was noted between the TCM and F-TCM groups. There was no significant difference in the apparent digestibility of CP, CF, ash, or NFE among all the groups (*p* > 0.05).

### 3.5. The Composition of Fecal Microbiota

At a cutoff level of 3%, no effect on the ACE and Chao richness estimators was observed in all groups (Figure 1). The Shannon diversity estimator in the F-TCM group was significantly increased (*p* < 0.05), while the Simpson index decreased compared with the control and TCM groups (*p* < 0.01).

The OTU distribution of the microbial communities of the different treatment groups had a certain degree of similarity and specificity. In order to understand the species differences, Venn diagrams were used to show the common and unique information among the different groups based on the OTU abundance information of the samples. As shown in Figure 2, 549 microbial species were common to all groups; however, 101 OTUs were unique in the control, and 136 and 146 OTUs were unique in the TCM and F-TCM groups, respectively.

At the phylum level, *Bacteroidetes* and *Firmicutes* were the predominant phyla in the fecal microbiota of piglets, with a total abundance of >95%, followed by the phyla *Actinobacteria* and *Proteobacteria* (Figure 3A,D). The abundance statistics of each taxon were visualized using Krona, the total profiling of the composition of microbial species is shown in Appendix A, and the composition of microbial species at the phylum level of *Firmicutes* (Appendix A), *Bacteroidetes* (Appendix A), and *Actinobacteria* (Appendix A) are presented in Appendix A.

At the genus level, in the phylum *Bacteroidetes*, 10 genera with a relative abundance of >1% were found to be dominant (Figure 3E). In the phylum *Firmicutes*, 17 genera with a relative abundance of >1% were found to be dominant (Figure 3F). Among these, compared with the control group, the relative abundance of *Erysipelotrichaceae*_UCG-004 and *Ruminococcaceae*_UCG-014 were increased, and the abundance of *Prevotellaceae*_UCG-003 and *Oscillospira* were decreased in the TCM and F-TCM groups (*p* < 0.05, Figure 4). Compared with the control and F-TCM groups, the abundance of *Acidaminococcus*, *Prevotella*_7, and *Megasphaera* were increased, while the abundance of *Bifidobacterium*, *Coprococcus*_1, *Lachnoclostridium*, *Prevotellaceae*_UCG-003, *Faecalibacterium*, and *Oscillibacter* were decreased in the TCM group (*p* < 0.05, Figure 4). Compared with the control and TCM groups, the abundance of *Coprococcus*_1, *Lachnoclostridium*, the *Lachnospiraceae*_FCS020 group, the *Prevotellaceae*_NK3B31 group, the *Ruminococcaceae*_UCG-014, and *Ruminococcaceae*_UCG-008 were increased, while the abundance of *Acidaminococcus*, *Prevotella*_7, and *Megasphaera* were decreased in the F-TCM group (*p* < 0.05, Figure 4). Compared with the TCM group, the relative abundance levels of *Coprococcus*_1, *Lachnoclostridium*, the *Lachnospiraceae*_FCS020 group, the *Prevotellaceae*_NK3B31 group, *Faecalibacterium*, *Oscillibacter*, and *Ruminococcaceae*_UCG-008 were increased, while the abundance of *Acidaminococcus*, *Bifidobacterium*, *Prevotella*_7, and *Megasphaera* were decreased in the F-TCM group (*p* < 0.05, Figure 4).

At the OTU level, compared with the control group, the relative abundance of *Ruminococcaceae*_UCG-014-related OTUs (OTU157 and OTU043), the *Subdoligranulum*-related OTU214, and the *Prevotellaceae*-related OTU292 were increased, while the abundance of the *Ruminococcaceae*_UCG-002-related OTU037 and the *Prevotella*_9-related OTU295 were decreased in the TCM and F-TCM groups (*p* < 0.05, Figure 5). Compared with the control and F-TCM groups, the abundance of the *Christensenellaceae*_R-7 group-related OTU210, the *Acidaminococcus*-related OTU112, and the *Prevotella*_7-related OTU007 were increased, while the abundance of the *Candidatus Soleaferrea*-related OTU133, the *Oscillibacter*-related OTU300, the *Oscillospira*-related OTUs (OTU242 and OTU041), the *Ruminiclostridium*_9-related OTU202, the *Prevotella*_1-related OTU069, the *Prevotella*_9-related OTUs (OTU131 and OTU057), and the *Prevotellaceae*_UCG-003-related OTU012 were decreased in the TCM group (*p* < 0.05, Figure 5). Compared with the TCM group, the abundance of the *Coprococcu*_1-related OTU315, the *Lachnoclostridium*-related OTU148, the *Candidatus Soleaferrea*-related OTU133, the *Oscillibacter*-related OTU300, the *Oscillospira*-related OTUs (OTU242 and OTU041), the *Ruminiclostridium*_9-related OTU202, the *Ruminococcaceae*_UCG-005-related OTU180, the *Ruminococcaceae*_UCG-014-related OTUs (OTU354, OTU184, OTU157, and OTU043), the *Prevotella*_1-related OTU069, and the *Prevotella*_9-related OTUs (OTU131 and OTU057) were increased, while the abundance of the *Ruminococcaceae*_UCG-002-related OTU037, the *Acidaminococcus*-related OTU112, and the *Prevotella*_7-related OTU007 were decreased in the F-TCM group (*p* < 0.05, Figure 5).

### 3.6. Correlation between Fecal Microbiota and Apparent Nutrient Digestibility

In order to investigate the relationship between the intestinal microbial community and apparent nutrient digestibility in piglets, the correlations were analyzed (Table 7 and Figure 6).

As shown in Table 7, the microbial richness estimators (ACE, Chao) were positively correlated with apparent DM digestibility (*p* < 0.05), and the microbial diversity indices (Shannon, Simpson) were positively correlated with apparent EE digestibility (*p* < 0.05).

At the genus level (Figure 6A), the relative abundance of the *Ruminococcaceae*_UCG-014 and *Lachnospiraceae*_FCS020 groups were negatively correlated with apparent DM digestibility (*p* < 0.05); the abundance of *Ruminococcaceae*_UCG-014 and *Lachnoclostridium* was negatively correlated with apparent EE digestibility (*p* < 0.05); and the abundance of *Oscillospira* was negatively correlated with apparent ash digestibility (*p* < 0.05).

At the OTU level (Figure 6B), the relative abundance of the *Ruminococcaceae*_UCG-014-related OTUs (OTU043 and OTU157) and the family *Prevotellaceae*-related OTU371 were positively correlated, while the abundance of the family *Ruminococcaceae*-related OTU037 and the *Prevotella*_9-related OTU295 was negatively correlated with the apparent DM digestibility (*p* < 0.05). The abundance of the *Subdoligranulum*-related OTU214 was negatively correlated with the apparent CP digestibility (*p* < 0.05); the abundance of the *Prevotella*_9-related OTU295, the family *Ruminococcaceae*-related OTUs (OTU103 and OTU165), and the family *Bacteroidale*_S24-7 group-related OTU113 was positively correlated with apparent EE digestibility, while the abundance of the *Ruminococcaceae*_UCG-014-related OTUs (OTU157, OTU354, and OTU180), the *Prevotellaceae*_NK3B31 group-related OTUs (OTU056, OTU065, OTU116, and OTU188), the *Prevotella*_1-related OTU069, the family *Ruminococcaceae*-related OTU292, and the *Lachnoclostridium*-related OTU148 was negatively correlated with the apparent EE digestibility (*p* < 0.05). The abundance of *Oscillospira*-related OTU041, *Prevotellaceae*_NK3B31 group-related OTU188, and family *Prevotellaceae*-related OTU292 were positively correlated with apparent CF digestibility (*p* < 0.05).

## 4. Discussion

With the implementation of the policy to remove AGPs in animal production in China, natural plants and their application as AGP substitutes have gained increasing interest in the research community because of their safety, efficiency, and availability [38]. TCMs are considered better alternatives for improving animal health and resisting infectious diseases. *Fructus mume* (“wumei” in Chinese) has long been used in China to treat chronic coughs, expectoration, ulceration, chronic diarrhea, and gastrointestinal diseases [39,40,41,42,43,44]. This medicinal effect is due to its antioxidant [44], antibacterial [45], and anti-inflammatory properties [43,44], and its protective ability against gastrointestinal diseases via the opsonization of intestinal commensal bacteria, as well as its ability to alleviate epithelial injury and inflammation [39]. *Scutellaria baicalensis* Georgi (“huangqin” in Chinese) is also an old and well-known component of TCM and is widely used for the treatment of bronchitis, hepatitis, tumors, and inflammatory diseases [46,47,48,49,50]. Numerous research studies have indicated that the therapeutic effects of *Scutellaria baicalensis* Georgi are due to its various pharmacological activities, including its antiangiogenic, anti-inflammatory, antimicrobial, immunoenhancing, and antioxidative effects [51,52,53,54]. Very little was found in the literature on the effects of dietary TCM in weaned piglets. Our study systematically investigated the data and aimed to ascertain the effects of these two TCM feed additives on the growth performance, apparent nutrient digestibility, and fecal microbiota of weaned piglets.

Prior studies have reported that fermented feed significantly increases the body weight and ADG of piglets [55], laying hen chicks [56], and geese [57]. Several reports have shown that TCM additives significantly improved the final BW, ADG, and FCR in lambs and hogs [12] and promoted growth performance in heat-stressed beef cattle, which was associated with better physiological status [11]. These findings are contrary to a previous study showing that dietary TCM led to greater feed intake but no significant differences in the final BW, ADG, or F:G ratio [58]. There were few reports on the effects of fermentation with TCM mixtures on the growth performance of weaned piglets. In our study, dietary supplementation with *Fructus mume* and *Scutellaria baicalensis* Georgi, fermented or not fermented, led to no significant differences in ADFI during the experiment period. The findings of the current study do not support previous research where the authors suggested that the inclusion of supplemental TCMs may improve pigs’ appetite [58]. The results of this study showed that the final BW and ADG increased, while the F:G ratio and diarrhea rate decreased in weaned piglets in the TCM and F-TCM groups, suggesting that *Fructus mume* and *Scutellaria baicalensis* Georgi supplementation in the diet improves growth performance, leading to greater weight gain and improved health [10,11,12]. However, fermentation with *Fructus mume* and *Scutellaria baicalensis* Georgi in the diet had no significant effect on the growth performance of piglets.

One interesting finding in our study was observed in the TCM and F-TCM groups, in which ADG was increased while ADFI was not significantly changed; considerably more work will need to be carried out to determine apparent nutrient digestibility in piglets. Previous studies have explored whether dietary TCM increases the apparent digestibility of CP in finishing pigs [59] and CP, Ca, P, and NDF in weaned piglets [60]. Dietary TCM has also been suggested to improve the apparent digestibility of OM, CP, ADF, Ca, and P in beef cattle, even under heat stress [10,11]. However, the findings of the current study do not support the previous research; as shown in Table 6, there was no significant difference in the apparent digestibility of CP or CF among the three groups. In addition, the partial substitution of fermented feed in the diet increased the apparent digestibility of CP and CF in growing-finishing pigs [61]; CP and EE in Xuefeng black-boned chickens [62]; and DM, CP, CF, NDF, and ADF in lactating dairy cows [63]. Data on fermented TCM feed are lacking; in our study, feed fermented with TCM increased the apparent digestibility of DM and EE compared with the non-fermented group (TCM group), suggesting that fermented feed with *Fructus mume* and *Scutellaria baicalensis* Georgi mixture increased the digestibility of DM and EE in weaned piglets.

Recently, studies have found that the gut microbiota may explain the therapeutic effects of TCM [6,64]. An increasing number of studies have investigated the interactions between the gut microbiota and TCM, suggesting that the gut microbiota can directly affect the absorption, metabolism, and pharmacological activity of TCM [2,3,65]. In our study, the composition of fecal microbiota in weaned piglets was analyzed. Dietary *Fructus mume* and *Scutellaria baicalensis* Georgi decreased the diversity but did not have a significant effect on the richness of fecal microbiota; this finding is inconsistent with that of Zou (2021), who reported that a Huangqin decoction (HQD) could increase the diversity of the intestinal microbiota of cholestatic mice [66]. This inconsistency may be due to the different species. The current study also found that feed fermented with these two TCMs improved the diversity of fecal microbiota in piglets. Previous results suggested that eight Chinese herbs (Chinese name: “jian ji san”) fermented with *Zygosaccharomyces rouxii* and their fermentation products increased the diversity of the foregut microbial community of broiler chickens [67], which is in agreement with our results. According to these data, we can infer that dietary *Fructus mume* and *Scutellaria baicalensis* Georgi affected the fecal microbial composition by altering its diversity in weaned piglets.

In our study, *Bacteroidetes* and *Firmicutes* were the predominant phyla in the fecal microbiota of piglets, similar to the findings of previous studies [28,68,69]. As mentioned in the literature review, the gut microbial mediation of the potential therapeutic mechanism of TCMs can be attributed to the production of short-chain fatty acids (SCFAs) [67,70], mostly acetic, propionic, and butyric acids, which play an important role in maintaining the intestinal health of pigs [70,71]. In Figure 4, there is a clear trend of the increased relative abundance of *Acidaminococcus*, *Prevotella*, and *Megasphaera* in the TCM group and a decreased abundance in the F-TCM group; all three genera are considered to be SCFA-producing bacteria [70,72], suggesting that the diet supplemented with *Fructus mume* and *Scutellaria baicalensis* Georgi increased the abundance of SCFA-producing bacteria in the hindgut, which may have promoted the intestinal health of the weaned piglets. However, the prefermentation of these two TCMs, dietary fibers, and other indigestible carbohydrates led to their degradation before being fed to the piglets [70], explaining the decreasing tendency of apparent CF digestibility and the abundance of *Acidaminococcus*, *Prevotella*, and *Megasphaera* in the F-TCM group. Another important finding in the current study was that the relative abundance of *Coprococcus*, *Lachnospiraceae*_FCS020 group, and *Oscillibacter* were increased in the F-TCM group; these genera have been identified as the most active and healthy microbiome constituents in the intestinal environment in healthy adult humans and animals [28,73,74,75]. Thus, these findings suggest that fermentation with *Fructus mume* and *Scutellaria baicalensis* Georgi results in the improved healthy intestinal flora, which, in turn, could favor the intestinal health of weaned piglets.

Another important objective of this study was to investigate the relationship between the gut microbial characteristics and the apparent nutrient digestibility in weaned piglets, so the correlation between the apparent nutrient digestibility and significant microbial genera and OTUs were further analyzed. Significant correlations between microbial diversity and apparent EE digestibility were observed, suggesting a potential link between changes in the intestinal flora and apparent EE digestibility. At the OTU level, the relative abundance of 10 OTUs that increased in the F-TCM group was positively correlated with apparent EE digestibility, while four OTUs for which the abundance was decreased in the F-TCM group were negatively correlated with apparent EE digestibility. Within these OTUs, most belong to the families *Ruminococcaceae* and *Prevotellaceae*, which are considered to be the core bacteria detected in 99% of fecal samples obtained from commercial swine worldwide [76]. Prior studies have shown that the abundance of *Ruminococcaceae* is negatively correlated with high-fat-diet (HFD)-induced obesity in a ripened pu-erh tea extract (PETe) intervention in mice [77], while the relative abundance of *Prevotellaceae* and *Prevotellaceae*_NK3B31_group was observed to increase through supplementation with pea seed coats (PSCs) and stachyose in mice and rats fed an HFD [78,79]. According to these findings, we can infer that the digestion and absorption of nutritional lipids in the diet were closely related to the changes in the *Ruminococcaceae* and *Prevotellaceae* families. Our correlation results are consistent with previous studies focusing on the gut microbiota and the apparent nutrient digestibility of grower pigs and sows [15,80] and support the possible relationship between the gut microbiota and the regulation of dietary nutrient utilization in piglets.

## 5. Conclusions

Dietary supplementation with *Fructus mume* and *Scutellaria baicalensis* Georgi in antibiotic-free feed improved the final BW and ADG, increased the abundance of SCFA-producing bacteria in the hindgut, and decreased the F:G ratio and diarrhea rate, yielding healthier weaned piglets that gained more weight. Fermentation with these TCMs enhanced the apparent digestibility of DM and EE and improved the healthy intestinal flora, which, in turn, could favor the intestinal health of weaned piglets. There was a significant correlation between the increased apparent EE digestibility in the TCM diets and the diversity of fecal microbiota. Dietary TCMs affect the fecal microbial composition by changing the abundance of certain genera belonging to the *Ruminococcaceae*, *Prevotellaceae*, and *Lachnospiraceae* families, which may further increase the apparent EE digestibility of weaned piglets. Nevertheless, our study cannot demonstrate causality, and further experimental studies are needed to address this.

## Figures and Tables

**Figure 1 animals-12-02418-f001:**
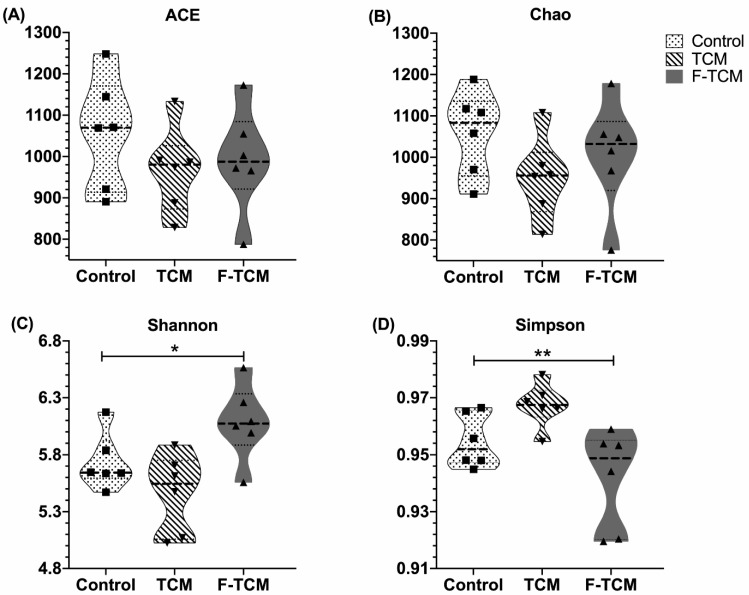
The richness and diversity of the fecal microbiota in weaned piglets. (**A**) ACE index, (**B**) Chao index, (**C**) Shannon index, (**D**) Simpson index; * indicate significant differences among three groups (*p* < 0.05), ** indicate significant differences among three groups (*p* < 0.01).

**Figure 2 animals-12-02418-f002:**
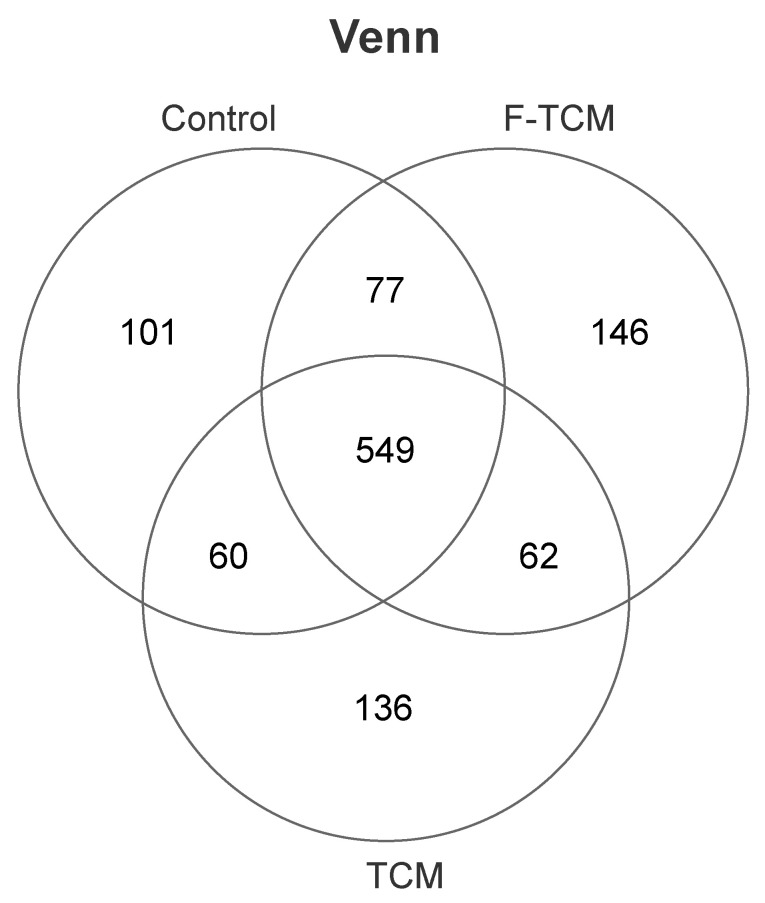
Venn diagram analysis of the OTUs among the treatment groups.

**Figure 3 animals-12-02418-f003:**
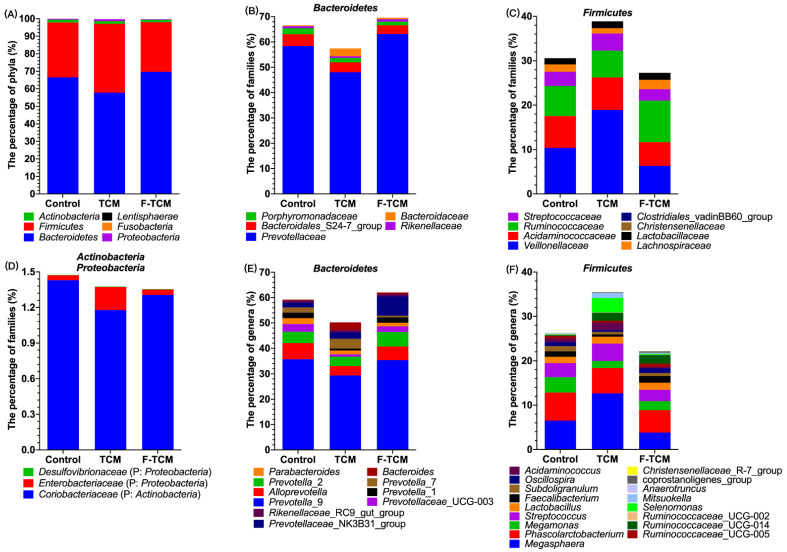
The composition and structure of the fecal microbiota in weaned piglets (relative abundance of more than 1%). The *Bacteroidetes* and *Firmicutes* phyla constituted approximately 95% of the identified sequences (**A**), followed by *Actinobacteria* and *Proteobacteria* as follows: five families (**B**) and 10 genera (**E**) in *Bacteroidetes*; eight families (**C**) and 17 genera (**F**) in *Firmicutes*; one family (**D**) in *Actinobacteria*; two families (**D**) in *Proteobacteria*. P: phylum.

**Figure 4 animals-12-02418-f004:**
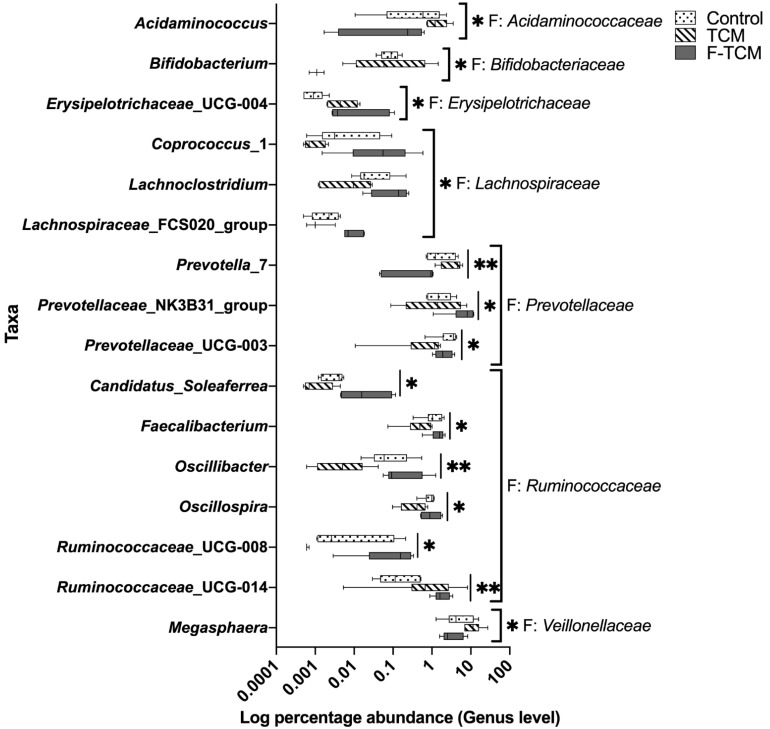
The relative abundance of the fecal microbiota of weaned piglets at the genus level. F: family. * indicate significant differences among three groups (*p* < 0.05), ** indicate significant differences among three groups (*p* < 0.01).

**Figure 5 animals-12-02418-f005:**
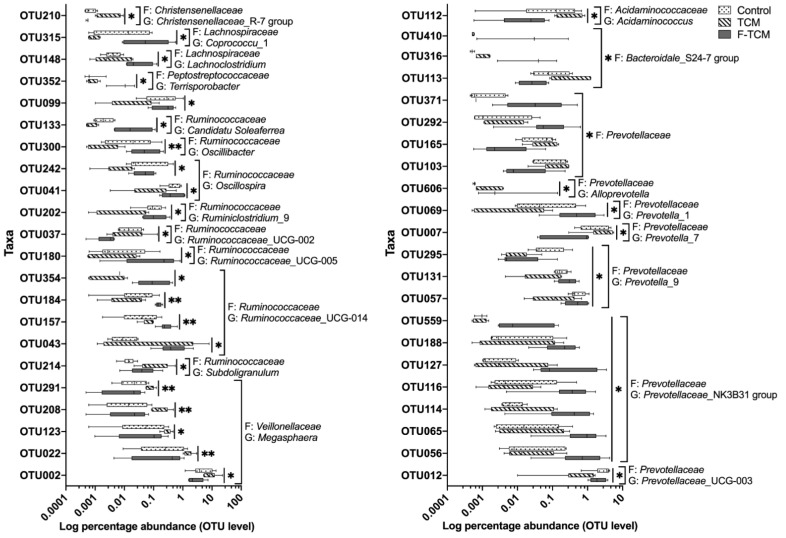
The relative abundance of the fecal microbiota of weaned piglets at the OTU level. F: family; G: genus. * indicate significant differences among three groups (*p* < 0.05), ** indicate significant differences among three groups (*p* < 0.01).

**Figure 6 animals-12-02418-f006:**
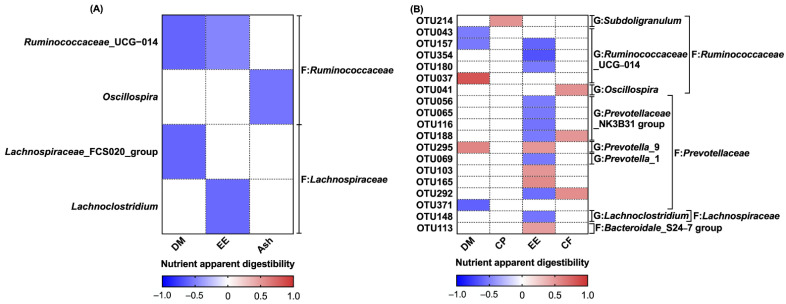
Correlation analysis of fecal microbiota (genus and OTU levels) and apparent nutrient digestibility: (**A**) genus level; (**B**) OTU level. The color indicates the Pearson coefficient distribution: red represents a positive correlation (*p* < 0.05), blue represents a negative correlation (*p* < 0.05), and white shows that the correlation was not significant (*p* > 0.05). F: family; G: genus.

**Table 1 animals-12-02418-t001:** The composition of fermented mixed feeds.

No.	Mixed Substrate, g				Fermentation Strain, g		
	Corn	Soybean	*Fructus mume*	*Scutellaria baicalensis* Georgi	*Bacillus subtilis*	*Lactobacillus*	*Saccharomyces*
Mixed substrates fermented with the fermentation strains							
1	222.75	74.25	3.00	-	0.60	0.60	0.60
2	218.25	72.75	9.00	-	0.60	0.60	0.60
3	202.50	67.50	30.00	-	0.60	0.60	0.60
4	222.75	74.25	-	3.00	0.60	0.60	0.60
5	218.25	72.75	-	9.00	0.60	0.60	0.60
6	202.50	67.50	-	30.00	0.60	0.60	0.60
7	225.00	75.00	-	-	0.60	0.60	0.60
Mixed substrates fermented without the fermentation strains							
8	222.75	74.25	3.00	-	-	-	-
9	218.25	72.75	9.00	-	-	-	-
10	202.50	67.50	30.00	-	-	-	-
11	222.75	74.25	-	3.00	-	-	-
12	218.25	72.75	-	9.00	-	-	-
13	202.50	67.50	-	30.00	-	-	-
14	225.00	75.00	-	-	-	-	-

**Table 2 animals-12-02418-t002:** Composition and nutrient levels of the experimental diets.

	Experimental Diets		
	Control	TCM	F-TCM
Ingredient (%), DM			
Corn	67.00	63.00	62.40
Soybean meal	25.00	25.00	25.00
Self-made premix ^a^	8.00	8.00	8.00
*Fructus mume*	-	1.00	1.00
*Scutellaria baicalensis* Georgi	-	3.00	3.00
*Bacillus subtilis*	-	-	0.20
*Lactobacillus*	-	-	0.20
*Saccharomyces*	-	-	0.20
Nutrition level (%) ^b^, DM			
Crude protein (CP)	22.97 ± 0.82	26.75 ± 0.73	27.85 ± 0.82
Ether extract (EE)	16.27 ± 0.58	15.48 ± 0.44	16.00 ± 0.24
Crude fiber (CF)	5.26 ± 1.57	6.29 ± 1.79	7.15 ± 0.46
Ash	5.40 ± 0.00	4.54 ± 0.02	5.36 ± 0.27
Nitrogen-free extract (NFE)	37.36 ± 2.99	44.21 ± 3.22	38.99 ± 4.59

^a^ Ingredients: fish meal, choline chloride, vitamin, mineral elements, *L*-lysine hydrochloride, calcium hydrophosphate, stone powder, sodium chloride, enzyme preparation, flavoring agent, and sweetening agent. No antibiotics were added. ^b^ Measured values (Mean ± SD).

**Table 3 animals-12-02418-t003:** The MICs of the water maceration extracts of eight TCMs for Escherichia coli and Salmonella.

TCMs	MICs, mg·mL^−1^	
	*Escherichia coli*	*Salmonella*
*Fructus mume*	25.00	30.00
*Scutellaria baicalensis* Georgi	35.00	40.00
*Rhizoma imperatae*	130.00	>250.00
*Paeoniae radix alba*	>250.00	>250.00
*Plantaginis semen*	>250.00	>250.00
*Eclipta prostrata*	>250.00	>250.00
*Fructus arctii*	>250.00	>250.00
*Portulaca oleracea* L.	>250.00	>250.00

**Table 4 animals-12-02418-t004:** The growth performance of weaned piglets (mean ± SD).

Measure	Experimental Diets			*p*-Value
	Control	TCM	F-TCM	
Initial BW, kg	6.69 ± 0.43	7.96 ± 0.89	7.99 ± 0.93	0.25
Final BW, kg	13.69 ± 0.80 ^b^	18.55 ± 1.06 ^a^	18.71 ± 2.04 ^a^	<0.05
ADG, g	259.26 ± 10.45 ^b^	392.32 ± 7.16 ^a^	397.19 ± 9.85 ^a^	<0.05
ADFI, g	535.67 ± 38.64	609.40 ± 14.50	654.44 ± 77.19	0.14
F:G, g/g	2.07 ± 0.01 ^a^	1.55 ± 0.03 ^b^	1.64 ± 0.08 ^b^	<0.05

^a,b^ Values within a row without a common superscript letter are significantly different (*p* < 0.05).

**Table 5 animals-12-02418-t005:** The DM of feces and the diarrhea rate of weaned piglets (mean ± SD).

Measure	Experimental Diets			*p*-Value
	Control	TCM	F-TCM	
Feces, DM %	60.16 ± 4.89 ^b^	63.15 ± 3.80 ^b^	67.34 ± 2.95 ^a^	<0.05
Diarrhea rate, %	22.22 ± 4.13 ^a^	12.96 ± 3.23 ^b^	7.66 ± 2.26 ^c^	<0.05

^a,b,c^ Values within a row without a common superscript letter are significantly different (*p* < 0.05).

**Table 6 animals-12-02418-t006:** The apparent nutrient digestibility of weaned piglets (mean ± SD).

Nutrient	Experimental Diets			*p*-Value
	Control	TCM	F-TCM	
DM, %	62.93 ± 5.12 ^b^	70.20 ± 0.99 ^a^	66.59 ± 4.66 ^a^	<0.05
CP, %	64.48 ± 5.67	68.86 ± 4.16	72.93 ± 6.63	0.26
EE, %	79.05 ± 3.75 ^b^	89.93 ± 6.48 ^a^	85.09 ± 4.34 ^a^	<0.05
CF, %	72.14 ± 4.23	86.57 ± 5.34	62.77 ± 2.65	0.18
Ash, %	85.82 ± 4.71	86.67 ± 3.34	87.21 ± 5.72	0.94
NFE, %	85.53 ± 5.46	68.25 ± 3.81	74.93 ± 3.37	0.54

^a,b^ Values within a row without a common superscript letter are significantly different (*p* < 0.05).

**Table 7 animals-12-02418-t007:** Correlations of fecal microbial richness and diversity estimators with apparent nutrient digestibility.

Apparent Digestibility, %	Correlation Coefficient			
	ACE	Chao	Shannon	Simpson
DM	0.03 *	0.05 *	0.53	0.16
CP	0.34	0.28	0.82	0.97
EE	0.89	0.71	0.03 *	0.02 *
CF	0.67	0.82	0.10	0.17
Ash	0.74	0.61	0.82	0.59
NFE	0.34	0.72	0.44	0.44

* Significant correlation (*p* < 0.05).

## Data Availability

Raw data collected and presented in this study are available on request from the corresponding author.

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
