# Peer review of "Effect of Dietary Fructus mume and Scutellaria baicalensis Georgi on the Fecal Microbiota and Its Correlation with Apparent Nutrient Digestibility in Weaned Piglets"

_animals, 2022, doi:10.3390/ani12182418_

Round 1

Reviewer 1 Report (Previous Reviewer 2)

This review report is based on author’s response.

Q1: Illumina sequencing platform that the author used to sequence the sample is still confusing. The line 187 (Hiseq) and line 206 (Miseq) shows us the different platform.

Q2: if the Venn results did not contribute to the present study and the author decided not to put the results in the manuscript, then what is the purpose of mentioning that this study has been done with Venn Analysis? If some analyses were performed and the results may not that much helpful for study, please, at least, put it the supplementary material.

Q3: The author’s answer is Silva, but the description (Line235-237) in the Manuscript shows a different story.

Q4: Krona results should also be presented to the reader.

Q5: The author mentioned that the Biomarker features in each group were screened by both Metastats and LEfSe software. Did Metastats and LEfSe give us the same biomarker?

Q7: the purpose of legend is to help others quickly understand and get what information that the figure presents. It is not like that the reader need to go back the manuscript to check what is the T meaning and what does the FT represent.

Q9: if those significantly different genera were analyzed by SPSS, then where is the results of Lefse and Metastats.

Q10: If those data were analyzed by Pearson, then why Line 251-252 mentioned “Correlation analysis was performed by Spearman's or Pearson's correlation tests.”?

Author Response

Thank you for the reviewers' comments concerning our manuscript entitled ”Effect of dietary Fructus mume and Scutellaria baicalensis Georgi on the fecal microbiota and its correlation with apparent nutrient digestibility in weaned piglets” (Manuscript ID: animals-1837679). Those comments are all valuable and very helpful for revising and improving our paper, as well as the important guiding significance to our researches. We have studied comments carefully and have made correction which we hope meet with approval. Revised portion are marked in red in the paper. Please see the attachment.

Reviewer 2 Report (New Reviewer)

Please see attached document with review comments 

Author Response

Thank you for the reviewers' comments concerning our manuscript entitled ”Effect of dietary Fructus mume and Scutellaria baicalensis Georgi on the fecal microbiota and its correlation with apparent nutrient digestibility in weaned piglets” (Manuscript ID: animals-1837679). Those comments are all valuable and very helpful for revising and improving our paper, as well as the important guiding significance to our researches. We have studied comments carefully and have made correction which we hope meet with approval. Revised portion are marked in red in the paper. Please see the attachment.

Reviewer 3 Report (New Reviewer)

In this manuscript entitled “Weaned piglet hindgut microbiome and apparent nutrient digestibility responses to antibiotic-free feed supplementation with Fructus mume and Scutellaria baicalensis Georgi”. This topic is extreme important nowadays because weanling period is vital for pig health and growth performance. Overall, this manuscript is well written. And I have several major concerns.

1. Table 2: in this table, the crude protein content of three diets is extreme high (26%-32%) and variable. Specifically, the crude protein content of T and FT group are both lower than basal diet. Consequently, it is difficult to eliminate the effect of different CP level on diarrhea rate and growth performance. Therefore, it may interfere with the effect of author’s supplementation. Please classify this.

2. The author conducted 16S sequencing using fecal samples, and then performed correlation analysis between fecal microbiota and nutrients digestibility. Generally, macro-nutrients are mainly digested in small intestine and fermentation occurs at large intestine. It is reasonable to perform analysis between fecal microbiota and nutrients digestibility because these two are totally unrelated. Please classify this.

Author Response

Thank you for the reviewers' comments concerning our manuscript entitled ”Effect of dietary Fructus mume and Scutellaria baicalensis Georgi on the fecal microbiota and its correlation with apparent nutrient digestibility in weaned piglets” (Manuscript ID: animals-1837679). Those comments are all valuable and very helpful for revising and improving our paper, as well as the important guiding significance to our researches. We have studied comments carefully and have made correction which we hope meet with approval. Revised portion are marked in red in the paper. Please see the attachment.

Round 2

Reviewer 1 Report (Previous Reviewer 2)

No further comments.

Author Response

We feel great thanks for your professional review work on our article entitled "Effect of dietary Fructus mume and Scutellaria baicalensis Georgi on the fecal microbiota and its correlation with apparent nutrient digestibility in weaned piglets" (Manuscript ID: animals-1837679). According to your nice suggestions, we have made correction which we hope meet with approval. Revised portion are marked in red in the paper. Please see the attachment.

Reviewer 2 Report (New Reviewer)

The authors have addressed the majority of my comments. 

Please add a sentence in the statistical analysis to clarify that initial body weight was used as a covariate in the analysis 

Author Response

We feel great thanks for your professional review work on our article entitled "Effect of dietary Fructus mume and Scutellaria baicalensis Georgi on the fecal microbiota and its correlation with apparent nutrient digestibility in weaned piglets" (Manuscript ID: animals-1837679). According to your nice suggestions, we have made correction which we hope meet with approval. Revised portion are marked in red in the paper. Please see the attachment.

This manuscript is a resubmission of an earlier submission. The following is a list of the peer review reports and author responses from that submission.

Round 1

Reviewer 1 Report

This study investigated the fecal microbiota in piglets when they were fed diets supplemented with Chinese herbal medicines with the aim of finding non-antibiotic feed supplements for pigs. Considering the increasing global concern due to rising AMR, the topic of this study is contemporary and important. The manuscript is adequately written. However, there are some major issues that need to be resolved before this manuscript could be considered for publication.

Title

Replace ‘hindgut microbiome’ with ‘faecal microbiota’

Introduction

Authors have provided some useful and relevant background information in this section. However, it is not clear what the motivation is behind conducting this study. Authors should provide more information to justify why this study was needed. For example, more information on the major chemical compounds in Chinese Herbal medicines, their antimicrobial activity, and any impact on the gut microbiota or nutritional performances in pigs or other animals should be provided such that the justification of this study is clear. In addition, authors should briefly justify the use of fermented herbal plants.

Line 53: use AGP instead of antibiotic growth promoters…change here and throughout

Line 55: …multiple classes of antibiotics…

Line 57-66: In addition to Cu, Zn has been heavily used in the UK and EU countries. Authors should include some background on the use of Zn in pig(let) diet.

Line 72-76: Authors should provide more background information about Chinese Herbal medicines and the major compounds, and their antimicrobial properties either in in vitro or in vivo conditions.

Line 77: …go through various challenges

Line 79: Not clear what authors mean by ‘unique intestinal microbiota population related to apparent nutrient digestibility’…should be clarified! Either use ‘microbiota’ or ‘microbial population’;

Line 86: ‘correlation’ rather than ‘relation’

Materials & Methods

What was the experimental unit? How was it decided that there should be 3 pens per diet or 54 pigs per diet or 18 pigs per pen?

Was there any diet adaptation period?

Line 93: What was the justification for using E. coli and Salmonella in MIC tests?

Sub-section 2.2: Without any prior information/background, it is difficult to understand why feed was fermented and used as one of the dietary treatments.

Line 138: Previously it was said that feed was fermented for 144 h, but here it says 72 h. Please clarify!

Line 147: Table 2: Ingredient % and Nutrition Level % are on fresh weight or dry weight basis? In addition, experimental diets should be as homogenous as possible in terms of their nutritional composition so that any unexpected bias in the results could be avoided. However, in the current study, CP% in the ‘control’ diet was higher compared to two experimental diets. Authors should explain how that variation in the diets could influence any results obtained in the current experiment.

Line 158-159: What was defined/considered as diarrhoea?

Line 178: The fecal samples…

Sub-section 2.8: Statistical analysis: How was the mean separated after ANOVA (for example, the data reported in Table 5)? Please mention that statistical significance was considered at P<0.05.

Line 222-223: Correlation analysis was done across all sample i.e. regardless of dietary treatment. Would not it be more important to know the relation between nutrient digestibility and microbial composition/structure within each dietary treatment? Did author consider RDA?

Results

Font size in each figure is too small (many times)

Line 232: …showed stronger antimicrobial effect…

Section 3.2 is useful, but it is not one of the objectives of this study. Authors should provide this information (Section 3.2 and Table 4) in a supplementary file (not in the main text).

Line 250: diarrhea rate is not a measure of growth performance

Line 251: delete ‘significantly’ here and elsewhere used for similar purpose; let the readers decide whether it is significant or not by looking at the numbers provided

Line 252: delete ‘Meanwhile’

Line 253: decreased compared to control?

Line 255: In Table 5, provide the P values rather than saying P < 0.05. What do the ±

values indicate, SD or SE? Mention it clearly as a footnote in each table

Line 256: In Table 6, provide the P values rather than saying P < 0.05.

Line 278-281: No mention of Fig 2D

Line 287-293: increased or decreased compared to?

Line 306-311: increased or decreased compared to?

Line 325-326: delete; repetition of what has been mentioned in the Materials & Methods

Discussion

Line 358: …prohibiting the use of AGP…

Line 359-360: …have gained increasing interest in research and as AGP substitutes…

Line 360: efficiency…what kind of efficiency? Antimicrobial efficiency?

Line 364-366: Rewrite to rectify the sentence structure and improve clarity

Line 358-376: This text is not a discussion of the findings of this study. It reads like a generic discussion or rather a piece of text on background information about Chinese Herbal medicine. Please delete!

Line 378-413: The discussion is a bit ambiguous. Improve the clarity!

Line 424: Different species of what? How about the difference in bioactive compounds present in different plant species?

Line 430-431: How did it influence the diversity? Explain or provide plausible explanation!

Line 454-473: The discussion is very confusing and does not lead to a clear key message. Please re-write to improve the clarity!

Conclusions

Line 480-482: Delete the objective statement from the conclusions section.

Line 483: Absorption…was it measured in this study? Avoid including unreasonable conclusions

Line 485: Causality in regard to?
